# Influence of conspiracy theories and distrust of community health volunteers on adherence to COVID-19 guidelines and vaccine uptake in Kenya

Edward Mugambi Ireri[1,2]*, Marion Wanjiku Mutugi[1], Jean-Benoît Falisse[3], James Mwirigi Mwitari[1], Lydia Kemunto Atambo[1]

**1** Amref International University, Nairobi, Kenya, **2** Smart Health Consultants Limited Company, Nairobi, Kenya, **3** School of Social and Political Science, University of Edinburgh, Edinburgh, United Kingdom

* ireri76@gmail.com

**Data Availability Statement:** The data underlying this article is publicly available at https://doi.org/10.5281/zenodo.6554640.

## Abstract

Public trust is key for compliance to government protocols in times of health mitigating COVID-19 measures and its vaccination initiative, and thus understanding factors related to community health volunteers (CHVs) trusting the government and conspiracy theories is vital during the COVID-19 pandemic. The success of universal health coverage in Kenya will benefit from the trust between the CHVs and the government through increased access and demand for health services. This cross-sectional study collected data between 25 May to 27 June 2021 and it involved CHVs sampled from four counties in Kenya. The sampling unit involved the database of all registered CHVs in the four counties, who had participated in the COVID-19 vaccine hesitancy study in Kenya. Mombasa and Nairobi (represented cosmopolitan urban counties). Kajiado represented a pastoralist rural county, while Trans-Nzoia represented an agrarian rural county. Probit regression model was the main analytical method which was performed using R script language version 4.1.2. COVID-19 conspiracy theories weakened generalised trust in government (adjOR = 0.487, 99% CI: 0.336–0.703). Banking on COVID-19 related trust in vaccination initiatives (adjOR = 3.569, 99% CI: 1.657–8.160), use of police enforcement (adjOR = 1.723, 99% CI: 1.264–2.354) and perceived risk of COVID-19 (adjOR = 2.890, 95% CI: 1.188–7.052) strengthened generalised trust in government. Targeted vaccination education and communication health promotion campaigns should fully involve CHVs. Strategies to counter COVID-19 conspiracy theories will promote adherence to COVID-19 mitigation measures and increase vaccine uptake.

## 1. Background

Public health responses to crises, including epidemics, typically relies on the idea that the public trusts the authorities. Low trust levels lead to low compliance with public health guidelines, as many studies have documented including research on Ebola outbreaks in Liberia [1] and

**Funding:** This research is part of an Epidemic Ethics/WHO initiative that FCDO/Wellcome Grant 214711/Z/18/Z has supported. WHO's specific grant number was 2020/1077878-0). The funders had no role in study design, data collection and analysis, decision to publish, or manuscript preparation. No authors received a salary from the funders.

**Competing interests:** The authors declare no conflict of interest exist.

the Democratic Republic of Congo [2]. 'Trust' is the psychological state of a trustor and can be defined as comprising the intention to accept one's vulnerability in a situation involving risk, based on positive expectations of the intentions or behaviour of the trustee [3, 4]. The trustees are the county and national governments in Kenya and COVID-19 is the risk factor. Studies of trust in the context of the COVID-19 pandemic [3, 5] have reported reduced government trust due to the perception of COVID-19 guidelines being too extreme, with conspiracy theory believers perceiving the measures as unnecessarily strict.

As demonstrated by Osur *et al.*, [6] CHVs play a significant role in building trust and convincing communities in Kenya to accept COVID-19 vaccination and treatment. It is therefore crucial to understand how they relate to official guidelines.

Conspiracy theories are particularly important when considering trust in government. A conspiracy theory is the belief that an individual(s) and/or organisation(s)–often a powerful actor, such as some government or billionaire–is secretly responsible for an event (COVID-19 in our case) or is hiding an important secret away from the public. In this regard, conspiracy theories, especially when involving governments, are markers of institutional mistrust and can undermine government pronouncements [7].

Distrust in the government, defined as the absence of trust in a situation of grave risk or deep doubt, and low levels of trust, in general, have been a problem during the COVID-19 pandemic, in Africa and elsewhere [8]. Indeed, it is widely accepted that when citizens trust their government, they are more likely to follow its guidelines about spatial distancing and other coronaviruses behavioural changes [9]. In their 2021 study covering 13 African countries, Adebisi *et al.* [8] explain that distrust in government fuelled by political corruption discouraged collaboration on government protocols, thus compromising the fight against the COVID-19 pandemic. Similar findings are reported in studies conducted in Nigeria [10] and DRC [11]. In Zambia [12], the population feared the politician would exploit COVID-19 to gain power or enrich themselves. Nigerians [13] believed that COVID-19 was created to allow more corruption. In Kenya [14], the perception of the management of the pandemic and lack of transparency in the use of COVID-19 donations/funds tainted the government's image and eroded public trust.

In environments marked by widespread distrust of the state and mainstream media as credible information sources, in Africa and elsewhere, rumour finds particular relevance and utility [15]; information conveyed unofficially by word of mouth is often taken by the general population and the elites to be more accurate than information shared by the government or other formal institutions [16]. Conspiracy theories also flourish in such environments, such as the myth that 5G technology was behind the COVID-19 pandemic that is reportedly widespread across Africa [17]. They negatively influence government trust and adherence to COVID-19 guidelines [18, 19]. The government's capacity to enforce public health measures do not only rely on citizens trusting and obeying their recommendations: coercion was used during the pandemic in many countries in Africa and elsewhere. It often led to abuse and an increase in rights violations of vulnerable groups [20]. To cite but a few examples: within the first days of South Africa's 21-day COVID-19 lockdown, numerous videos emerged allegedly depicting police and soldiers kicking, slapping, whipping and shooting lockdown violators [21]; in the coastal city of Mombasa, the Kenyan police officers were filmed beating people waiting for a passenger ferry, as well as journalists covering the events two hours before the curfew [22]; and in Zimbabwe, the Association of Doctors for Human Rights reported harassment and assault of ordinary citizens by state security agents during the COVID-19 induced national lockdown [23]. Such policing of the response to COVID-19, Jones explains [24], had a long-lasting impact on the legitimacy and police-community relationships far beyond the pandemic. Writing on Nigeria, Aborisade [25] noted that the police 'militarised' response

engendered deeper divides between the police and communities than previously existed due to police aggression.

A study in Zambia [26] stated that refusal of COVID-19 vaccination would reflect a lower perceived risk related to a higher perceived risk of contracting the infection. Perceptions of risk partly map on what is known of the actual risk factors; for instance Kim *et al.*, [27] in South Africa found that the likelihood of sensing the danger of COVID-19 infection increased mostly among adults. A comparative survey among sub-Saharan Africans and Africans living in the diaspora found that the youth had lower risk perception scores than the older respondents [28]. Other factors also come into play [29] as the same study finds that those employed and those with higher education levels had significantly higher risk perceptions scores than the other respondents.

The data from the government of Kenya, shows that the country had been struggling to attain its target of the number of adults who have received the full dose of the COVID-19 vaccine. Equally, Kenyans had been struggling to adhere to COVID-19 guidelines. These issues pose a challenge to the government as it embarks on Universal Health Coverage (UHC) among Kenyans. In 2018, President Kenyatta declared UHC as one of the "Big Four" strategic pillars which would see major policy and administrative reforms in the medical sector, thus ensuring every Kenyan has access to quality and affordable medical coverage by 2022. CHVs are the link between the formal health system and the grassroots on matters health as they can be used to increase access and demand for health services (COVID-19 guidelines and vaccine uptake in the context of this paper). They act as the 'gatekeepers' to health in their community and understanding their perceptions of COVID-19 guidelines and vaccine uptake will play a major role in the uptake of these health services among Kenyans.

In this study, we focus on a key agent of the COVID-19 response in Kenya, the CHVs. This study investigates if trust in the context of the COVID-19 pandemic is perceived differently under the devolved government in Kenya.

In this context, the study investigates CHVs trust in COVID-19 messages (promoting pro-health behaviours such as mitigation measures and vaccination) issued by the national and the devolved county governments in Kenya.

Thus, the objective of the current study is to explore the influence of COVID-19 conspiracy theories, trust at the three levels of government, and experience of police enforcement of COVID-19 measures and perceived risks on adherence to COVID-19 guidelines and vaccine uptake in Kenya among CHVs. Fig 1 shows the conceptual framework used in the study.

## 2. Materials and methods

This study was conducted 15 months after COVID-19 measures were first announced and implemented in Kenya. Due to the challenging economic measures, the study incorporated a question on self-confirmation as a CHVs. Certainly, it was possible that out of frustration, some would have quit their roles as CHVs. The study involved CHVs from four counties: Nairobi, Mombasa, Kajiado and Trans Nzoia.

This cross-sectional study collected data between 25 May –27 June 2021. It involved all registered community health volunteers (CHVs) who had participated in the COVID-19 vaccine hesitancy study [6], whose database was used as the sampling unit.

The sample size of 372 CHVs is drawn from a total national population of 2,786 CHVs (from Nairobi 1,386, Mombasa 539, Kajiado 365 and Trans Nzoia 496) for whom we had complete email addresses and mobile phone contacts. A confidence level of 0.05, an alpha of 0.05 and a population proportion of 0.5, was used to determine a sample size of 338, of which 10% was added for attrition.

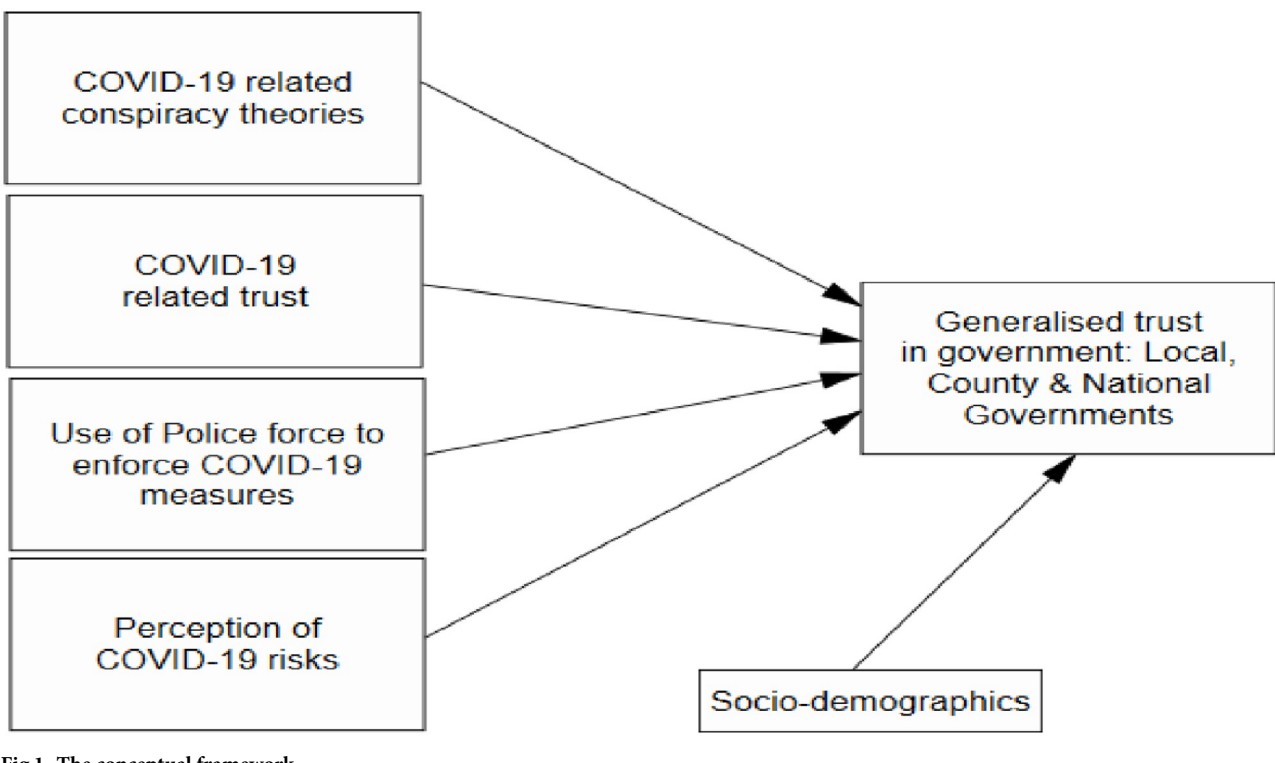

**Fig 1. The conceptual framework.**

The questionnaire, adapted from Vinck *et al's*., study [2] (S1 Appendix), was uploaded on the survey monkey online platform and the link was emailed to each CHVs. Observing that the response rate was very low, the university verified bulk SMS platform from Safaricom was used to send individual survey links to the mobile phones of the CHVs, leading to a tenfold increase in the response rate. The study questionnaire had three dependent variables, that is, trust at the three levels of the government: local, County and National governments. The study had four categories of independent variables: COVID-19 conspiracy theories (7 items), COVID-19 related trust (4 items), police enforcement of COVID-19 measures (1 item) and perceived risks associated with COVID-19 (5 items) adapted from Vinck *et al's*., study [2], which were also the covariate in the logit and probit regression analysis. The significant social demographics used in the study included: Monthly income, age, household head, gender, pay cut and place of worship were the covariates variables.

Community health workers are community members with basic training to promote health or carry out limited healthcare services but they are generally not healthcare professionals [30]. In Kenya, community health workers are volunteers who do not officially draw a monthly income, although in some instances, they are paid a stipend as a motivating factor and to compensate for their time. And hence, for the sake of this study, we stick to the phrase community health volunteers (CHVs). One of the main roles of CHVs in Kenya is to support child health and immunization, counselling on immunisation schedules, mobilising communities during immunisation days, identifying and refering children for immunisation, tracing and referring defaulters, and assisting in immunisation campaigns [31]. Therefore, it is expected that CHVs will play a major role in convincing communities in Kenya to accept COVID-19 guidelines and vaccine uptake.

Most of the CHVs ($n = 260;86.7\%$) had a secondary school level of education. Only 40 (13.3%) had only a primary school level of education. A closer look at the data shows that most of CHVs ($n = 38;95\%$) with a primary education came from urban places (in slum areas) while only 2(5%) came from rural places (Table 1). The CHVs attended a three days virtual training on data collection using survey monkey.

**Table 1. Descriptive statistics.**

| Variable | Characteristic | Frequencies |
|---|---|---|
| Age | Youth ≤ 35 years | 130 (43.3%) |
| | Adult ≥ 36 | 170 (56.7%) |
| Gender | Male | 88 (29.3%) |
| | Female | 212 (70.7%) |
| Pay cut | No | 68 (22.7%) |
| | Yes | 232 (77.3%) |
| Monthly income | Under KES 25,000 | 277 (92.3%) |
| | Over KES 25,000 | 23 (7.7%) |
| Household members | ≤ 4 persons | 131 (43.7%) |
| | ≥ 5 persons | 169 (56.3%) |
| Place of worship | Roman Catholic | 95 (31.7%) |
| | Protestant | 68 (22.7%) |
| | Evangelical Churches | 43 (14.3%) |
| | African Instituted Churches | 11 (3.7%) |
| | Other Christians | 60 (20%) |
| | Islam | 23 (7.7%) |
| Marital status | Never married | 95 (31.7%) |
| | Married | 123 (41%) |
| | Living with a partner | 16 (5.3%) |
| | Divorced | 43 (14.3%) |
| | Widowed | 23 (7.7%) |
| Level of education | Primary | 40 (13.3%) |
| | Secondary | 140 (46.7%) |
| | University undergraduate | 53 (17.7%) |
| | University postgraduate | 3 (1.0%) |
| | Other tertiary colleges | 64 (21.3%) |
| News media for receiving COVID-19 updates | Television | 171 (57%) |
| | Radio | 42 (14%) |
| | Internet | 36 (12%) |
| | CHVs | 21 (7%) |
| | Others | 30 (10%) |
| Area of residence | Urban city | 119 (39.7%) |
| | Urban municipality | 34 (11.3%) |
| | Urban county | 76 (23.5%) |
| | Urban township | 32 (10.7%) |
| | Rural town | 27 (9%) |
| | Rural countryside | 12 (4%) |
| Job loss due to COVID-19 measures | Yes | 251 (83.7%) |
| | No | 49 (16.3%) |
| Member of any association | | |

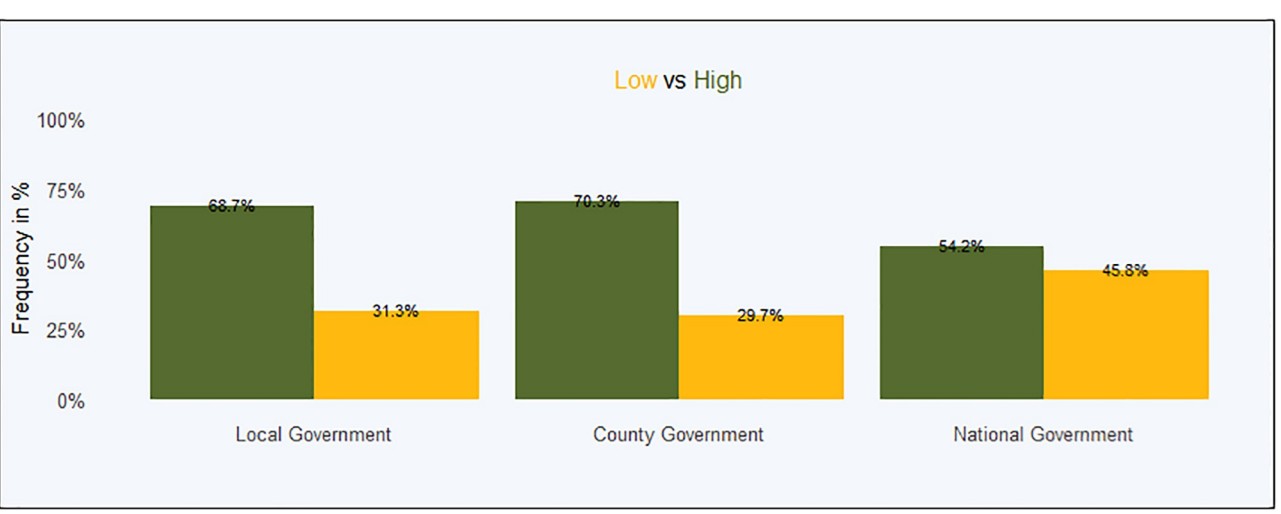

**Fig 2. Overall trust in government.**

The preliminary analysis on measures of associations between the dependent and the independent variables focused on chi-square statistics which were computed using *summarytools* package in R [32]. All the independent social demographics that had no significant association with the dependent variables (Trust at the three levels of the government), were dropped from further analysis.

The inferential statistics were evaluated and interpreted using the confidence level at 95% as a baseline. The models include a wide range of controls that are found in S1 Appendix, we do not report the value of each of these covariates among our main results. The covariates that are consistently associated with higher trust in government, across models, are the following: monthly income, age, household head, gender, pay cut, and place of worship. Trust in the three government levels was the main dependent variable and its summary statistics is presented in Fig 2.

Logistic and probit regressions were the main analytical methods, first because the outcome variables were re-coded in binary format. Secondly, logistic and probit regressions require fewer analytical assumptions as compared to multiple linear regression. Thirdly, logistic and probit regressions were used to determine associated factors without controlling for specific variables.

Hierarchical multiple Logistic and Probit regression were computed using *R* script language version 4.1.2 [33]. The interpretation focussed on crude odds and adjusted odds ratio. The publication-ready tables were plotted using the *Stargazer* package [34]; the pseudo-R-squared *fmsb* package generated the variance of the independent variables explained on the dependent variable [35]; and data manipulation was performed using the *dplyr* package [36].

The Amref Ethical and Scientific Review Committee approved the research protocol (ESRC P962/2021), and the National Commission for Science, Technology and Innovation issued the research license (NACOSTI-P-21-10063). The anonymous survey monkey, the key informant interviews with public officials in their official capacity, and the desk review were granted an exemption by the WHO ERC secretariat (CERC.0091). Written Informed consent was sought from all study participants. The study was undertaken by a team of researchers affiliated with Smart Health Consultants Limited Company, Amref International University, and the University of Edinburgh.

## 3. Main results

### 3.1 Socio-demographics

The total number of complete interview responses was 445. The demographic frequencies were computed using Dominic [32] summary tools package (S1 Appendix). Table 1 shows the descriptive statistics of the demographics used in the study.

Age, monthly income and household head were re-coded to binary variables to allow running the logit and probit regressions. Marital status, level of education, news media, area of residence, job loss, and being a member of any association or group had no significant association with the three dependent variables, and thus were dropped from further investigations on both the logit and probit regression models.

### 3.2 Inferential statistics

**3.2.1 COVID-19 conspiracy theories.** Sixty-one and one tenth percent of the respondent believed in at least one conspiracy theory among seven assertion that we had identified as widespread in the population (median 1). The most widespread are "COVID-19 was created by foreigners" (44.1%), "the threat of COVID-19 was exaggerated in Kenya" (30%), and "COVID-19 was eradicated in Kenya" (11.9%). Table 2 presents summary statistics on conspiracy theories.

Fig 3 presents overall statistics on conspiracy theories.

Table 3 presents the main associations between trust in government and the seven conspiracy theories. As with other results in this paper, we restrict the main findings to self-confirmed CHVs, the appendix presents the findings for those who could not self-confirm as a robustness check. Two theories, that COVID-19 was exaggerated and that it was eradicated are significantly associated with distrust in government at all levels. Additionally, the idea that Covid is not real was associated with generalised trust in county government (adj OR = 0.580, 95% CI: 0.343–0.981). These findings are robust to using both the Logit model and running the specifications on the sample of non-self-confirmed CHVs (S2 Appendix).

**3.2.2 COVID-19 related trust.** The frequencies of COVID-19-related trust are presented in Table 4, while the overall frequencies are shown in Fig 4.

**Table 2. COVID-19 conspiracy theories among self-confirmed CHVs n = 300 (frequencies).**

| Independent Variable | Response | Local government | | County Government | | National Government | |
|---|---|---|---|---|---|---|---|
| | | Low | High | Low | High | Low | High |
| (1). COVID-19 is real | Disagree | 81 (27%) | 189 (63%) | 75 (25%) | 195 (65%) | 67 (22.3%) | 203 (67.7%) |
| | Agree | 13 (4.3%) | 17 (5.7%) | 14 (4.7%) | 16 (5.3%) | 11 (3.7%) | 19 (6.3%) |
| (2). COVID-19 was eradicated in Kenya | Disagree | 82 (27.3%) | 175 (58.3%) | 78 (26%) | 179 (59.7%) | 70 (23.3%) | 187 (62.3%) |
| | Agree | 12 (4%) | 31 (10.3%) | 11 (3.7%) | 32 (10.7%) | 8 (2.7%) | 35 (11.7%) |
| (3). COVID-19 has never existed in my country | Disagree | 84 (28%) | 197 (65.7%) | 82 (27.3%) | 199 (66.3%) | 73 (24.3%) | 208 (69.3%) |
| | Agree | 10 (3.3%) | 9 (3%) | 7 (2.3%) | 12 (4%) | 5 (1.7%) | 14 (4.7%) |
| (4). The threat of COVID-19 was exaggerated in Kenya | Disagree | 53 (17.7%) | 160 (53.3%) | 46 (15.3%) | 167 (55.7%) | 41 (13.7%) | 172 (57.3%) |
| | Agree | 41 (13.7%) | 46 (15.3%) | 43 (14.3%) | 44 (14.7%) | 37 (12.3%) | 50 (16.7%) |
| (5). COVID-19 was created by the government of Kenya | Disagree | 88 (29.3%) | 203 (67.7%) | 84 (28%) | 207 (69%) | 75 (25%) | 216 (72%) |
| | Agree | 6 (2%) | 3 (1%) | 5 (1.7%) | 4 (1.3%) | 3 (1%) | 6 (2%) |
| (6). COVID-19 was created by foreigners | Disagree | 39 (13%) | 116 (38.7%) | 35 (11.7%) | 120 (40%) | 30 (10%) | 125 (41.7%) |
| | Agree | 55 (18.3%) | 90 (30%) | 54 (18%) | 91 (30.3%) | 48 (16%) | 97 (32.3%) |
| (7). COVID-19 is a white man disease | Disagree | 79 (26.3%) | 185 (61.7%) | 72 (24%) | 192 (64%) | 63 (21%) | 201 (67%) |
| | Agree | 15 (5%) | 21 (7%) | 17 (5.7%) | 19 (6.3%) | 15 (5%) | 21 (7%) |

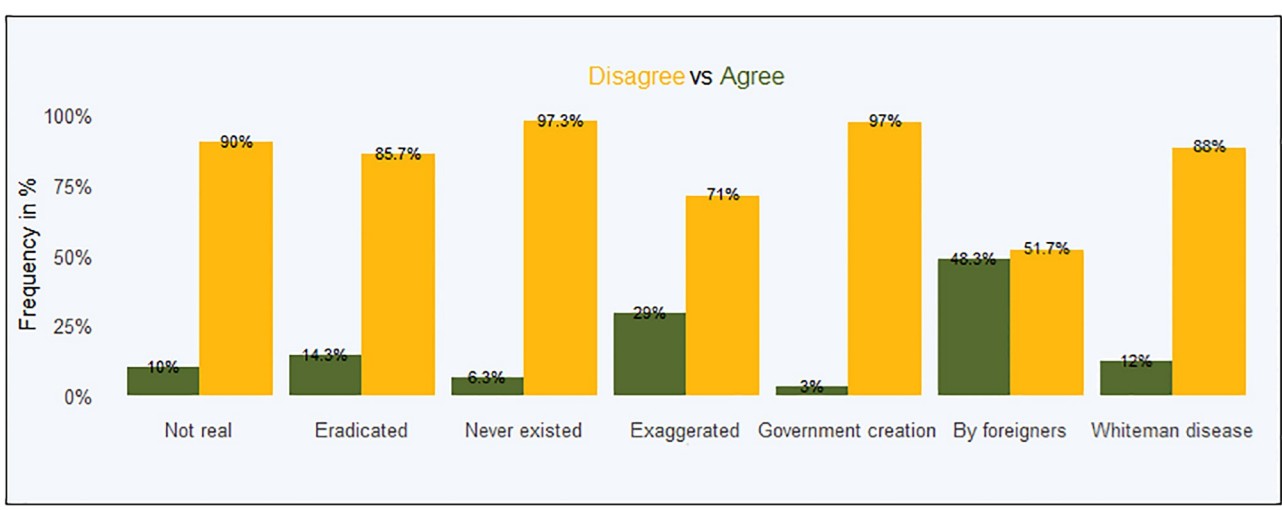

**Fig 3. Overall COVID-19 conspiracy theories.**

We now turn to trust in the COVID-19 response. As Table 5 shows, there is a very strong association between general trust in government at all levels and trust in the COVID-19 response as well as trust in the COVID-19 vaccination initiative. Interestingly, trust in parties or elements that are not directly related to the government, such as health professionals or the vaccine, are not correlated with trust in the government. The non-self confirmed model is in S3 Appendix. Monthly income (adjOR = 0.755, 95% CI: 0.612–0.932) and age (adjOR = 1.721, 95% CI: 1.038–2.855), were significantly associated with trust in the COVID-19 response. The

**Table 3. COVID-19 conspiracy theories among self-confirmed CHVs (probit model).**

| Independent variables | Dependent variable: Generalised trust in government | | |
|---|---|---|---|
| | Local Government Beta & *p*-value | County Government Beta & *p*-value | National Government Beta & *p*-value |
| (1). COVID-19 is real | -0.259; $p = 0.0331$ | -0.545; $p = 0.042$** | -0.439; $p = 0.109$ |
| (2). COVID-19 was eradicated in Kenya | 0.572; $p = 0.025$** | 0.652; $p = 0.013$** | 0.712; $p = 0.010$*** |
| (3). COVID-19 has never existed in my country | -0.462; $p = 0.169$ | 0.033; $p = 0.927$ | 0.137; $p = 0.709$ |
| (4). The threat of COVID-19 was exaggerated in Kenya | -0.619; $p = 0.001$*** | -0.792; $p = 0.00002$*** | -0.720; $p = 0.0002$*** |
| (5). COVID-19 was created by the government of Kenya | -0.657; $p = 0.167$ | -0.387; $p = 0.409$ | 0.026; $p = 0.957$ |
| (6). COVID-19 was created by foreigners | -0.154; $p = 0.372$ | -0.167; $p = 0.346$ | -0.208; $p = 0.252$ |
| (7). COVID-19 is a white man disease | -0.049; $p = 0.846$ | -0.279; $p = 0.269$ | -0.282; $p = 0.267$ |
| Constant | 0.768; $p = 0.000$*** | 0.897; $p = 0.000$*** | 0.975; $p = 0.000$****** |
| Observations ($n = 300$) | 300 | 300 | 300 |
| Log Likelihood | -173.758 | -164.605 | -157.247 |
| Akaike Inf. Crit. | 368.516 | 345.21 | 330.495 |

Notes

*p<0.1;

**p<0.05;

***p<0.01.

Beta = Beta estimates

**Table 4. COVID-19-related trust among self-confirmed CHVs n = 300 (frequencies).**

| Independent Variable | Response | Local government | | County Government | | National Government | |
|---|---|---|---|---|---|---|---|
| | | **Low** | **High** | **Low** | **High** | **Low** | **High** |
| (1). Trust government COVID-19 response | Low | 44 (14.7%) | 22 (7.3%) | 48 (16%) | 18 (6%) | 48 (16%) | 18 (6%) |
| | High | 50 (16.7%) | 184 (61.3%) | 41 (13.7%) | 193 (64.3%) | 30 (10%) | 204 (68%) |
| (2). Trust health professionals regarding COVID-19 response | Low | 26 (8.7%) | 10 (3.3%) | 26 (8.7%) | 10 (3.3%) | 22 (7.3%) | 14 (4.7%) |
| | High | 68 (22.7%) | 196 (65.3%) | 63 (21%) | 201 (67%) | 56 (18.7%) | 208 (69.3%) |
| (3). Trust COVID-19 vaccines administered in Kenya | Low | 43 (14.3%) | 21 (7%) | 47 (15.7%) | 17 (5.7%) | 42 (14%) | 22 (7.3%) |
| | High | 51 (17%) | 185 (61.7%) | 42 (14%) | 194 (64.7%) | 36 (12%) | 200 (66.7%) |
| (4). Trust the government's COVID-19 vaccination initiative | Low | 44 (14.7%) | 22 (7.3%) | 47 (15.7%) | 19 (6.3%) | 45 (15%) | 21 (7%) |
| | High | 50 (16.7%) | 184 (61.3%) | 42 (14%) | 192 (64%) | 33 (11%) | 201 (67%) |

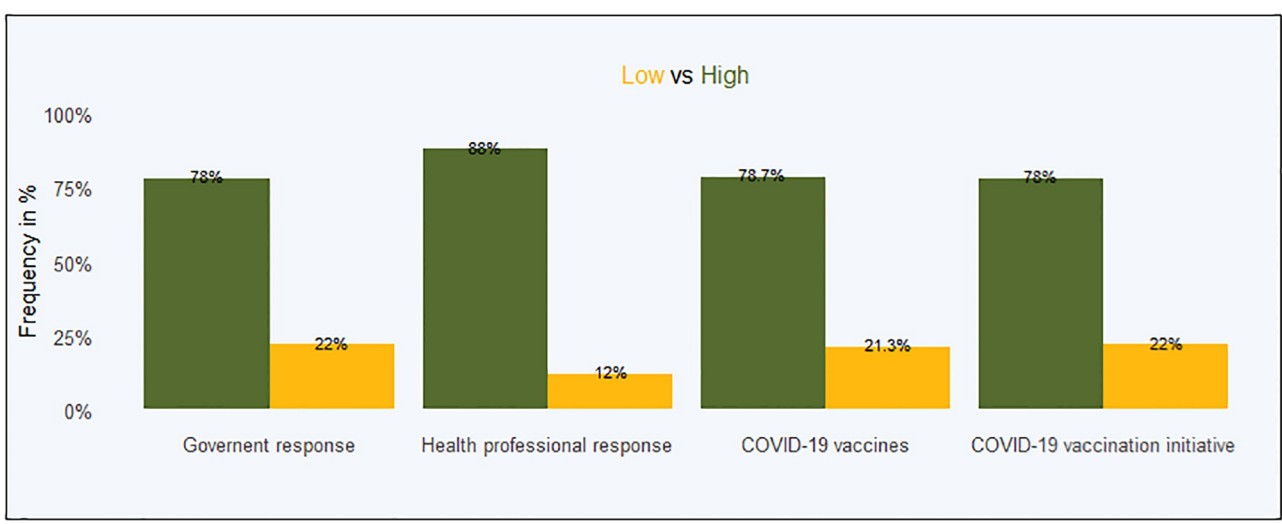

**Fig 4. Overall COVID-19 related trust issues.**

**Table 5. COVID-19-related trust among self-confirmed CHVs (probit model).**

| Independent variables | Dependent variable: Generalised trust in government | | |
|---|---|---|---|
| | **Local Government Beta & _p_-value** | **County Government Beta & _p_-value** | **National Government Beta & _p_-value** |
| (1). Trust the government's COVID-19 response | 0.707; $p = 0.002$*** | 1.071; $p = 0.00001$*** | 1.501; $p = 0.000$*** |
| (2). Trust health professionals regarding COVID-19 response | 0.374; $p = 0.212$ | 0.079; $p = 0.804$ | -0.594; $p = 0.079$* |
| (3). Trust COVID-19 vaccines administered in Kenya | 0.602; $p = 0.031$** | 0.880; $p = 0.003$*** | 0.467; $p = 0.117$ |
| (4). Trust the government's COVID-19 vaccination initiative | 0.376; $p = 0.198$ | 0.453; $p = 0.131$ | 0.910; $p = 0.004$*** |
| Constant | -1.111; $p = 0.00002$*** | -1.335; $p = 0.00001$*** | -0.925; $p = 0.0004$*** |
| Observations (_n_) | 300 | 300 | 300 |
| Log Likelihood | -151.753 | -130.172 | -113.276 |
| Akaike Inf. Crit. | 313.506 | 270.343 | 326.552 |

Notes

*p<0.1;

**p<0.05;

***p<0.01.

Beta = Beta estimates

**Table 6. Perception of police enforcement of COVID-19 measures among self-confirmed CHVs n = 300 (frequencies).**

| Independent Variable | Response | Local government | | County Government | | National Government | |
|---|---|---|---|---|---|---|---|
| | | Low | High | Low | High | Low | High |
| (1). Police enforcement | Not necessary | 55 (18.3%) | 72 (24%) | 54 (18%) | 73 (24.3%) | 46 (15.3%) | 81 (27%) |
| | Necessary | 39 (13%) | 134 (44.7%) | 35 (11.7%) | 138 (46%) | 32 (10.7%) | 141 (47%) |

binary logistic regression model is statistically insignificant by the Hosmer and Lemeshow Test $\chi^2$ (8) = 3.23; $p$ = 0.919, meaning that the model is a good fit.

Household head (adjOR = 0.409, 95% CI: 0.213–0.785) and gender (adjOR = 2.780, 95% CI: 1.526–5.064), were significantly associated with trust in the COVID-19 vaccination initiative. The binary logistic regression model is statistically insignificant by the Hosmer and Lemeshow Test $\chi^2$ (8) = 10.96; $p$ = 0.204, meaning that the model is a good fit.

Monthly income (adjOR = 0.755, 95% CI: 0.612–0.932) and age (adjOR = 1.721, 95% CI: 1.038–2.855), were significantly associated with trust in the government COVID-19 response. The binary logistic regression model is statistically insignificant by the Hosmer and Lemeshow Test $\chi^2$ (8) = 3.23; $p$ = 0.919, meaning that the model is a good fit. Household head (adjOR = 0.409, 95% CI: 0.213–0.785) and gender (adjOR = 2.780, 95% CI: 1.526–5.064), were significantly associated with trust in the COVID-19 vaccination initiative. The binary logistic regression model is statistically insignificant by the Hosmer and Lemeshow Test $\chi^2$ (8) = 10.96; $p$ = 0.204, meaning that the model is a good fit.

**3.2.3 The use of national police service in enforcing COVID-19 measures.** The summary statistics on the enforcement of COVID-19 measures are presented in Table 6.

Table 7 shows that trust in the use of police to enforce COVID-19 measures significantly correlated with generalised trust in local government (adjOR = 1.796, 99% CI: 1.329–2.431), county government (adjOR = 1.906, 99% CI: 1.406–2.589) and national government (adjOR = 1.723, 99% CI: 1.264–2.354). These findings are robust to using the Logit model but not to running the specifications on the sample of non-self-confirmed CHVs (S4 Appendix). Pay cut (adjOR = 1.961, 95% CI: 1.206–3.189) and place of worship (adjOR = 0.903, 95% CI: 0.821–0.993), were significantly associated with trust in the use of police to enforce COVID-19 measures. The binary logistic regression model is statistically insignificant by the Hosmer and Lemeshow Test $\chi^2$ (8) = 9.67; $p$ = 0.289, meaning that the model is a good fit.

**Table 7. Perception of police enforcement of COVID-19 measures among self-confirmed CHVs (Probit model).**

| Independent variables | Dependent variable: Generalised trust in government | | |
|---|---|---|---|
| | Local Government Beta & $p$-value | County Government Beta & $p$-value | National Government Beta & $p$-value |
| (1). Police enforcement of COVID-19 measures | 0.585; $p$ = 0.0002*** | 0.645; $p$ = 0.00004*** | 0.544; $p$ = 0.001*** |
| Constant | 0.169; $p$ = 0.132 | 0.189; $p$ = 0.092* | 0.353; $p$ = 0.002*** |
| Observations ($n$) | 300 | 300 | 300 |
| Log Likelihood | -179.218 | -173.724 | -165.985 |
| Akaike Inf. Crit. | 362.437 | 351.449 | 335.969 |

Notes

*p<0.1;

**p<0.05;

***p<0.01.

Beta = Beta estimates

**Table 8. COVID-19 perceived risks among self-confirmed CHVs n = 300 (frequencies).**

| Independent Variable | Response | Local government | | County Government | | National Government | |
|---|---|---|---|---|---|---|---|
| | | Low | High | Low | High | Low | High |
| (1). Vulnerability to COVID-19 infection | Disagree | 46 (15.3%) | 105 (35%) | 41 (13.7%) | 110 (36.7%) | 43 (14.3%) | 108 (36%) |
| | Agree | 48 (16%) | 101 (33.7%) | 48 (16%) | 101 (33.7%) | 35 (11.7%) | 114 (38%) |
| (2). COVID-19 poses a serious threat | Disagree | 28 (9.3%) | 39 (13%) | 25 (8.3%) | 42 (14%) | 25 (8.3%) | 42 (14%) |
| | Agree | 66 (22%) | 167 (55.7%) | 64 (21.3%) | 169 (56.3%) | 53 (17.7%) | 180 (60%) |
| (3). Precautions lower the risks of contracting COVID-19 | Disagree | 11 (3.7%) | 14 (4.7%) | 12 (4%) | 13 (4.3%) | 13 (4.3%) | 12 (4%) |
| | Agree | 83 (27.7%) | 192 (64%) | 77 (25.7%) | 198 (66%) | 65 (21.7%) | 210 (70%) |
| (4). Strong immune system against the common cold | Disagree | 83 (27.7%) | 197 (65.7%) | 81 (27%) | 199 (66.3%) | 71 (23.7%) | 209 (69.7%) |
| | Agree | 11 (3.7%) | 9 (3%) | 8 (2.7%) | 12 (4%) | 7 (2.3%) | 13 (4.3%) |
| (5). Likelihood of contracting COVID-19 | Unlikely | 38 (12.7%) | 59 (19.7%) | 35 (11.7%) | 62 (20.7%) | 30 (10%) | 67 (22.3%) |
| | Likely | 56 (18.7%) | 147 (49%) | 54 (18%) | 149 (49.7%) | 48 (16%) | 155 (51.7%) |

Pay cut (adjOR = 1.961, 95% CI: 1.206–3.189) and place of worship (adjOR = 0.903, 95% CI: 0.821–0.993), were significantly associated with trust in the use of police to enforce COVID-19 measures. The binary logistic regression model is statistically insignificant by the Hosmer and Lemeshow Test $\chi^2$ (8) = 9.67; $p$ = 0.289, meaning that the model is a good fit.

**3.2.4 COVID-19 perception of risk.** The frequencies of COVID-19 perception on perceived risks among self-confirmed CHVs are presented in Table 8 and Fig 5.

Finally, the probit model among self-confirmed CHVs (Table 9) shows that the perception that COVID-19 poses a serious threat significantly strengthened generalised trust in local government (adjOR = 1.495, 95%CI:1.021–2.188). Other variables of perceived risk matter: the perception that having a strong immune system against common cold could protect one from contracting COVID-19 significantly weakened generalised trust in local government (adjOR = 0.527, 95%CI:0.295–0.938). Precautions on COVID-19 significantly strengthened generalised trust in the national government (adjOR = 2.890, 95% CI: 1.188–7.052). The non-self confirmed model is shown in S5 Appendix. Income (adjOR = 1.304, 95% CI: 1.054–1.613) and household head (adjOR = 0.477, 95% CI: 0.280–0.812), were significantly associated

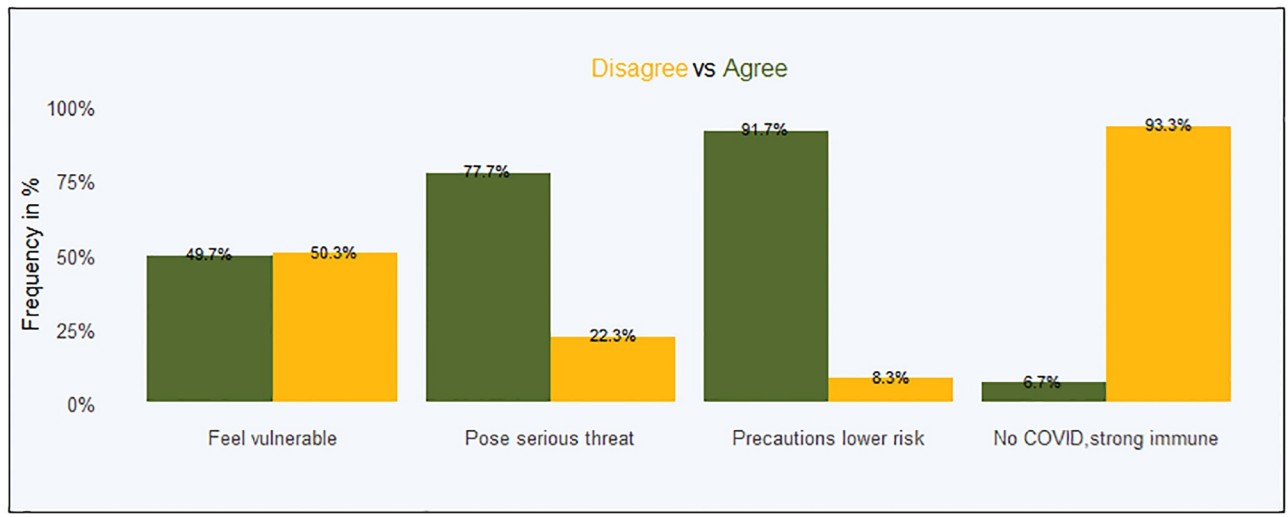

**Fig 5. Overall COVID-19 perceived risks.**

with vulnerability to COVID-19 infection. The binary logistic regression model is statistically insignificant by the Hosmer and Lemeshow Test $\chi^2$ (8) = 6.86; $p$ = 0.552, meaning that the model is a good fit.

Age (adjOR = 0.403, 95% CI: 0.166–0.979), was statistically associated with precautions lower the risk of contracting COVID-19. The binary logistic regression model is statistically insignificant by the Hosmer and Lemeshow Test $\chi^2$ (8) = 0.675; $p$ = 1.000, meaning that the model is a good fit.

Income (adjOR = 1.304, 95% CI: 1.054–1.613) and household head (adjOR = 0.477, 95% CI: 0.280–0.812), were significantly associated with vulnerability to COVID-19 infection. The binary logistic regression model is statistically insignificant by the Hosmer and Lemeshow Test $\chi^2$ (8) = 6.86; $p$ = 0.552, meaning that the model is a good fit. Age (OR = 0.403, 95% CI: 0.166–0.979), was statistically associated with precautions lower the risk of contracting COVID-19. The binary logistic regression model is statistically insignificant by the Hosmer and Lemeshow Test $\chi^2$ (8) = 0.675; $p$ = 1.000, meaning that the model is a good fit.

## 4. Discussion

We found a strong association between trust in government and trust in the COVID-19 response, including the police response, as well as a strong negative association between believing some conspiracy theories about COVID-19 and trust in government. These findings are consistent with COVID-19 studies on political corruption in Nigeria [10] and DRC [11], fear of political exploitation in Zambia [12], exaggeration of COVID as a government creation in Nigeria [13], a scum to exploit public funds in Kenya [14], whipping and shooting COVID-19 lockdown violators in South Africa [21], militarized police response in Nigeria [25] and slapping by the police in Kenya [22], all which had a strong negative association in trust in government.

Our data does not allow to qualify the nature of the relationship between trust in government and trust in the response, the police, or belief in conspiracy theories. The most likely is

**Table 9. COVID-19 perceived risks among self-confirmed CHVs (Probit model).**

| Independent variables | Dependent variable: Generalised trust in government | | |
| --- | --- | --- | --- |
| | Local Government Beta & $p$-value | County Government Beta & $p$-value | National Government Beta & $p$-value |
| (1). Vulnerability to COVID-19 infection | -0.239; $p$ = 0.155 | -0.325; $p$ = 0.055* | -0.006; $p$ = 0.971 |
| (2). COVID-19 poses a serious threat | 0.402; $p$ = 0.039** | 0.278; $p$ = 0.158 | 0.304; $p$ = 0.125 |
| (3). Precautions lower the risks of contracting COVID-19 | 0.260; $p$ = 0.354 | 0.483; $p$ = 0.084* | 0.654; $p$ = 0.020** |
| (4). Strong immune system against the common cold | -0.640; $p$ = 0.033** | -0.298; $p$ = 0.324 | -0.352; $p$ = 0.250 |
| (5). Likelihood of contracting COVID-19 | 0.296; $p$ = 0.082* | 0.278; $p$ = 0.105 | 0.124; $p$ = 0.481 |
| Constant | -0.085; $p$ = 0.757 | -0.116; $p$ = 0.672 | -0.231; $p$ = 0.402 |
| Observations ($n$) | 300 | 300 | 300 |
| Log Likelihood | -178.939 | -176.456 | -165.402 |
| Akaike Inf. Crit. | 369.879 | 364.913 | 342.805 |

Notes

*p<0.1;

**p<0.05;

***p<0.01.

Beta = Beta estimates

that this relationship goes both ways: the pre-pandemic levels of trust affect whether someone will take a positive view of the action of government or police or succumb to conspiracy theories. Conversely, bad experiences of the action of government or police or a repeated exposure to conspiracy theories may then erode trust in governments. COVID-19 guidelines such as lockdown, restrictions on travel bans, closure of businesses, job losses and pay cut, coupled with high unemployment rates, might have contributed to negative feelings. The COVID-19 pandemic made life difficult for most Kenyans, especially those who live in cities and have many mouths to feed, and our results suggest that trust in government is indeed less in towns and among large families. This finding is similar to Gagliardone et al., [19] study on the role played by COVID-19 myths among adult females in Africa. Despite the (few) government support measures, the main narrative in Kenya was one of the COVID-19 crisis being an opportunity for corruption and creating so-called "COVID-19 billionaires". The harshness of the measures, and the involvement of the measures also made numerous people belief that COVID-19 Kenya's pandemic was somehow exaggerated. This could have further fueled the negative attitude towards trusting the government.

The current study did not investigate the impact of political campaigns on trust. However, it is important to mention that uncontrolled political campaigns could have negatively impacted the confidence of CHVs in the generalised trust in government. Televisions were broadcasting directives on COVID-19 and political rally campaigns simultaneously. Probably most Kenyans were confused on what to trust. This could easily have made Kenyans link the political campaigns with the exaggeration of the COVID-19 pandemic. Equally, the do not care attitude by the politicians disobeying the government orders on public gatherings weakened the levels of trust. Politicians have a significant role to play in society as they have hardcore supporters who can be easily swayed by public utterances and the behaviour of their leaders. COVID-19 guidelines had four social-behavioural aspects: social distancing, handwashing and coughing etiquette, handshaking, and wearing face (mouth) masks. An open violation of the four measures by the political elites in their 2022 political campaigns was observed [37]. Social distancing was not/ and has not been adhered to in the campaign trails. The irony is that it was being enforced in places of worship and public transport [37]. The COVID-19 crisis was also a time for the revival of old and the creation of new conspiracy theories, which might have led to low levels of trust in the national government. The president received the AstraZeneca jab while his deputy went for a sputnik jab. This was then interpreted to mean the two powers were reading from different books, and thus it was a tough choice among Kenyans in trusting the government-initiated vaccines. Thus, a low turnout for the second dose vaccination in Kenya.

The conspiracy that COVID-19 was created by foreigners could easily damage the generalised trust at all levels of government, which would harm the awareness on the adherence to COVID-19 guidelines, including vaccination. A good example was the allegation that the Nyanza region immigration officials had issued 25 foreigners from India to work at Kibos sugar and allied industries in Siaya county. Five of them tested positive for the delta variant of COVID-19 [38]. Such allegation could fuel the belief that foreigners created COVID-19.

Trust in the COVID-19 response was significantly correlated with trust of all levels of government. The trust by the urban dwellers reflects in response to the COVID-19 vaccination campaign. The odds in trust increased gradually from the local to the national government. This is attributed to the government's early warning measures when COVID-19 was declared a pandemic by the WHO in March 2020. The government of Kenya has also been providing daily COVID-19 updates on both mass media and social media. CHVs earning less monthly had 1.3 times lesser odds of trusting the government's COVID-19 response compared with those earning more, while older CHVs ($\geq$ 36 years old) had 1.7 times greater odds of trusting

the government's COVID-19 response compared with the younger CHVs ($\leq 35$ years old). Lockdown measures and the closure of businesses meant that people would spend much time at home without a stable source of income. And thus, those earning less were against some measures which interfered with their source of income.

Trust in the COVID-19 vaccine was significant at local and county governments. Low income, family head, job loss, pay cut, and urban residency can be easily linked with tough economic times during the pandemic, making the CHVs trust the vaccine would open up the economy quickly, leading to normal life. The ministry of health data in the public domain shows that most counties considered more urban had registered high numbers of vaccinated persons compared with those classified as rural.

The relationship between trust in the COVID-19 vaccination and trust in county government was significantly strengthened by demographics that influence economics. For instance, monthly income. This study reported CHVs earning less monthly had 1.3 times lesser odds of trusting the government's COVID-19 response compared with those earning more. The trust in the vaccination initiative can be associated with the quicker opening of the economy and normal lifestyle. This finding is in agreement with Osur *et al's* [6] study, that reported a significant association between intention to accept COVID-19 vaccines and trust in MoH (national government) decisions on COVID-19 vaccination. As of the time of writing this paper, Kenya's government was struggling to vaccinate 10 million adult Kenyans by the end of December 2021. Equally, the uptake of the second dose was low. This cannot be associated with distrust in the vaccination initiative but with the conspiracy theories discussed in this paper. Not being a household head had 2.4 times lesser odds of trusting the government's COVID-19 vaccination initiative compared with those who were household heads, while male CHVs had 2.8 times greater odds of trusting the government's COVID-19 vaccination initiatives compared with the female CHVs. This study did not probe why more males were likely to trust the government's COVID-19 vaccination initiative compared with females.

The relationship between police enforcement of COVID-19 measures and generalised trust at all levels of governments was significantly strengthened by socio-demographics (pay cut and places of worship. The National Police Service was ordered to enforce the public health act, the outlined matatu regulations and quarantine regulations, and support the National Government Administration Officers (NGAOs) under the Ministry of Interior to enforce various drafted provisions COVID-19 measures and directives. However, several complaints of extortion and brutality by police enforcing the night curfew orders were reported [39], including abuses [40]. There were also cases where police arrests [41] were made amidst complaints from public members that several entertainment joints in Nairobi were operating illegally past curfew hours [42], not forgetting the curfew killings by the police [43]. Despite these negative issues, there was also the good side of the police. For example, a traffic policewoman won the hearts of many Kenyans after helping a woman and her son during the first night of the countrywide curfew [44], helping a lost drunkard during curfew [45], assisting a woman and her kid to reach the hospital during curfew [46], and helping locals fetch water to beat curfew [47]. With the challenging economic conditions and high crime rate, as reported by the Kenya diaspora messenger, it was a welcome for the police to patrol at night and keep criminals at bay [48], and thus the positive sentiments. The situation for Kenya needs to be interpreted with caution because the data only investigated CHVs.

The CHVs who had experienced pay cut or had someone in their household who had a pay cut had 1.9 times greater odds of trusting the use of police force to enforce COVID-19 measures, while (place of worship) not being a roman catholic had 1.1 times lesser odds of trusting the use of police force to enforce COVID-19 measures compared with those who were from other religions. Pay cut means loss of a steady source of income. It would be possible that those

CHVs who had experienced pay cut were likely to accept the role played by the police in enforcing COVID-19 measures with the hope that more people would abide by the measures, bringing down the positivity rates and thus the government lifting measures which harmed the economy. The Catholics hold weekly small prayer groups at their homes within their communities, unlike other denominations. Maybe, they were fine with the use of police enforcing the COVID-19 measures. Equally, they were already used to the regular presence of the police manning the church entrances every Sunday as a security measure against potential terrorist threats which maybe was not the case with other places of worship.

The CHVs who earned less income had 1.3 times greater odds of perceiving they were vulnerable to COVID-19 infection compared with those who earned more while not being the head of a household had 2.1 times lesser odds of perceiving they were vulnerable to COVID-19 compared with those who were household heads. When the COVID-19 pandemic was announced in Kenya, there was a shopping panic which included the purchase of hand sanitisers, masks and even oxygen cylinders by the rich. The best hospitals were full of admissions by the rich in society, and thus, a sense of hopelessness might have set in among those who felt that in case they contract COVID their hopes of survival were at risk.

A younger CHV ($\leq$ 35 years old) had 2.5 times lesser odds of perceiving precaution lowers the risk of contracting COVID-19 compared with an older CHV ($\geq$ 36 years old). Before the pandemic, life was tough among the youths and the pandemic came to worsen the situation. Probably it was tough for the youths to balance COVID-19 precautionary measures and earning a daily income. Lest we forget, most CHVs have no stable income and they depend on stipends which are paid based on the project activities in their communities. Thus, the youth would go a step further to search for the elusive income without adhering hundred percent to COVID-19 precautionary measures. Probably, the older people knew they were much more at risk from COVID-19.

Socio-demographics played a role in strengthening the relationship on perception that taking "*precaution lowers the risk of contracting COVID-19*", and "*influence of COVID-19 poses a severe threat*", at the county and the national government. These findings are in agreement with [27–29] studies that reported age, education level, marital status had significant influence on the perceived risks of COVID-19. The *perception that a "strong immune system against the common cold would protect one from COVID-19"* significantly weakened generalised trust in the county government. The effect was influenced by the family heads, low income, and pay cuts. All these indicate the need for targeted vaccination education and communication effort through CHVs.

Low adherence to COVID-19 measures and vaccine acceptance among the CHVs has highlighted the role of covariates associated with an economic crisis during the pandemic: job loss, pay cuts, low income, large families to feed etc.

The current study shows that trust among the CHVs can directly influence the uptake of COVID-19 vaccines and adherence to COVID-19 guidelines and much of the distrust can be attributed to dissatisfied CHVs. Community health volunteers are community members who are not under a formal payroll. Most of them have their side hustles (small jobs) and depend on tokens paid to them while assisting in projects running in their areas of residence.

Lack of adequate resources and remuneration was one of the limitations that led to the responses in the current study. Inadequate resources and remunerations may hinder the CHVs from serving their clients well and especially attending to households within their areas of jurisdiction at odd hours. They need to be facilitated with resources to improve their performances and increase trust in the ministry of health messaging [6] and they need further training to enable them effectively perform their roles.

The Kenyan government is known to allocate enormous budgets for employing short-term personnel to help in conducting population-based surveys and elections. If the same can be extended to the CHVs, the government will register high vaccine uptake and adherence to COVID-19 measures at present and in future pandemics.

Therefore, both the Counties and the National government must take care of the welfare of the CHVs through the provision of allowances and personal protective equipment (PPEs) to enable them to operate smoothly as they discharge their duties. It is worth noting the CHVs model in Kenya is weak and their remunerations have not been integrated within the devolved government. This poses a challenge not only to the COVID-19 vaccination initiative but also to the UHC. The County and the National government must address the remuneration challenges if this workforce is to be tapped to fully support policies and health interventions. The CHVs will play a critical role in the increased access and demand for health services as the government roll out the UHC to all Kenyans by 2030. Equally, to increase the trust and visibility of CHVs among community members, branding CHVs for ease of identification can increase the uptake of health services. The branding can be done through the provision of official identification cards, branded clothes such as T-shirts, jackets, caps, and PPEs as well as work kits.

## 5. Conclusion

Community health volunteers have a significant positive perception of COVID-19 related trust, police enforcement of COVID-19 measures, and COVID-19 perceived risks on generalised trust in government. The current study reports monthly income, age, household head, gender, pay cut, and place of worship as covariates that are consistently associated with higher trust in government on matters regarding COVID-19 measures and the uptake of its vaccines. Conspiracy theories weaken generalised trust in government. Neutralising COVID-19 conspiracy theories and promoting COVID-19 mitigation measures come hand in hand with fostering trust in government.

## Supporting information

**S1 Appendix. The questionnaire and its frequency summary.**
(DOCX)

**S2 Appendix. COVID-19 conspiracy theories among non-self-confirmed CHVs.**
(DOCX)

**S3 Appendix. COVID-19-related trust among non-self-confirmed CHVs.**
(DOCX)

**S4 Appendix. Perception of police enforcement of COVID-19 measures among non-self-confirmed CHVs.**
(DOCX)

**S5 Appendix. COVID-19 perceived risks among non-self-confirmed CHVs.**
(DOCX)

## Author Contributions

**Conceptualization:** Edward Mugambi Ireri, Marion Wanjiku Mutugi.

**Data curation:** Edward Mugambi Ireri.

**Formal analysis:** Edward Mugambi Ireri.

**Funding acquisition:** Marion Wanjiku Mutugi.

**Investigation:** Edward Mugambi Ireri, Lydia Kemunto Atambo.

**Methodology:** Edward Mugambi Ireri, Marion Wanjiku Mutugi, Jean-Benoît Falisse, James Mwirigi Mwitari.

**Project administration:** Edward Mugambi Ireri, Marion Wanjiku Mutugi, Lydia Kemunto Atambo.

**Software:** Edward Mugambi Ireri.

**Supervision:** Lydia Kemunto Atambo.

**Validation:** Jean-Benoît Falisse, James Mwirigi Mwitari.

**Visualization:** Edward Mugambi Ireri.

**Writing – original draft:** Edward Mugambi Ireri.

**Writing – review & editing:** Marion Wanjiku Mutugi, Jean-Benoît Falisse, James Mwirigi Mwitari, Lydia Kemunto Atambo.

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
