## [Decision Letter · Decision Letter 0]

21 Jun 2022

PGPH-D-22-00823

Influence of conspiracy theories and distrust of community health volunteers on adherence to COVID-19 guidelines and vaccine uptake in Kenya

Dear Dr. Edward Mugambi Ireri

Thank you for submitting your manuscript to PLOS Global Public Health. After careful consideration, we feel that it has merit but does not fully meet PLOS Global Public Health’s publication criteria as it currently stands. Therefore, we invite you to submit a revised version of the manuscript that addresses the points raised during the review process.

Please submit your revised manuscript by . If you will need more time than this to complete your revisions, please reply to this message or contact the journal office at globalpubhealth@plos.org. Please include the following items when submitting your revised manuscript:

We look forward to receiving your revised manuscript.

Kind regards,

Nusrat Homaira

Academic Editor

Journal Requirements:

State the initials, alongside each funding source, of each author to receive each grant.

2. Please ensure that the funders and grant numbers match between the Financial Disclosure field and the Funding Information tab in your submission form. Note that the funders must be provided in the same order in both places as well.

3. Please update your Competing Interests statement. If you have no competing interests to declare, please state: “The authors have declared that no competing interests exist.”

4. In the online submission form, you indicated that “The data underlying this article will be shared upon reasonable request to the corresponding author.”. All PLOS journals now require all data underlying the findings described in their manuscript to be freely available to other researchers, either 1. In a public repository, 2. Within the manuscript itself, or 3. Uploaded as supplementary information.

5. All main tables must be in an editable format for typesetting. Please include your tables in your main text as an editable table and not an image.

6. Please provide separate figure files in .tif or .eps format and remove any figures embedded in your manuscript file. Please also ensure that all files are under our size limit of 10MB.

Additional Editor Comments (if provided):

Reviewers' comments:

Reviewer's Responses to Questions

**Comments to the Author**

1. Does this manuscript meet PLOS Global Public Health’s publication criteria? Is the manuscript technically sound, and do the data support the conclusions? The manuscript must describe methodologically and ethically rigorous research with conclusions that are appropriately drawn based on the data presented.

Reviewer #1: Yes

Reviewer #2: Partly

2. Has the statistical analysis been performed appropriately and rigorously?

Reviewer #1: Yes

Reviewer #2: No

3. Have the authors made all data underlying the findings in their manuscript fully available (please refer to the Data Availability Statement at the start of the manuscript PDF file)?

Reviewer #1: No

Reviewer #2: Yes

4. Is the manuscript presented in an intelligible fashion and written in standard English?

Reviewer #1: No

Reviewer #2: No

5. Review Comments to the Author

Reviewer #1: The paper, “Influence of conspiracy theories and distrust of community health volunteers on adherence to COVID-19 guidelines and vaccine uptake in Kenya” how conspiracy theories and distrust influence community volunteers to adhere the COVID-19 guidelines and vaccine uptake in Kenya. The paper might be considered for publication, but I also think the paper needs major revisions to avoid repetition, to improve precision and clarity. In the abstract, the rationale for this study should be strengthen along with what additional knowledge this paper is going to add to the existing literature. The timeline of the study is missing in the abstract. CHVs, should be defined before using them in the abstract. The abstract could be benefitted by discussing some quantitative data.

Specific comments

The objective of the paper should clearly be mentioned in the last paragraph of the introduction. The literature review should be shortened and merged with the introduction. Follow the IMRAD.

Who conducted the study is missing.

The questionnaire, adapted from ---you should mention the study here not the number reference.

Hierarchical multiple logistic and probit regressions were done using (R script)3---You need to mention the version of the package. Rscript is not a software.

In the result section, the data has been presented in way that makes it difficult to follow.

The discussion should be started with a summary finding followed by supporting evidence from prior literature. The subheadings in the discussion can be avoided.

The tables can be provided in word.

You need to provide operational definitions of conspiracy theories and distrust.

Reviewer #2: Mutugi et al (2022) presented the influence of conspiracy theories and distrust of community health volunteers on adherence to COVID-19 guidelines and vaccines uptake in Kenya. While the findings are useful in national and regional scale, the article is written poorly, data are not presented convincing way, and conclusion are superficial and thus this article needs substantial improvement for being acceptance in Plos GPH.

Overall comments

1. The author did not present any data in the abstract and without doing so the author wrote a conclusion (“COVID-19 conspiracy theories weakened generalised trust in government”.) and recommendations (“Targeted vaccination education and communication health promotion campaigns should fully involve CHVs. Strategies to counter COVID-19 conspiracy theories will promote adherence to COVID-19 mitigation measures and increase vaccine uptake”).

Anyone reading the abstract of the article should have their own judgement to make evaluate the conclusion. If any data are not presented – the readers will not be able to judge the importance of the findings and credibility of the works. I suggest that the revised draft manuscript include raw data from the study needed to write the conclusions and recommendation.

2. The author included a section named “Literature review”. There is no such section recommended from this journal and not from any other journal that we know. This is a scientific manuscript NOT academic thesis. Please merge important information from literature review into Background section.

3. Methods: Please elaborate the statistical methods (3.5). Described how you have selected the variables for the model and how you have dropped them. I suggest you avoid sub-headings.

4. Results:

Please present the raw data i.e the value for n, N and % for each variable. This comment is applicable for tables 1-8 where applicable. The titles of most of the tables are vague – please be specific about the contents of the tables including duration, and locations of the study.

In the current format of the table I find it very difficult to interpret the findings. What does the “Porbit local value” -0.259 for “COVID-19 in not real” means (Table 1, row 1)? The p-value 0.331. What does this mean? Does these mean the differences among the “Porbit local” participants? I am struggling to understand the values here. Please revise the table here and elsewhere in the manuscript.

5. Discussion: please drop the sub-headings in the discussion section.

6. Conclusion: The author concluded that “Low income, family heads, job loss, pay cut, household members, urban residency, television news, gender, age, member of an association, education, religion, and marital status are potential covariates that influence generalised trust in government on matters COVID-19 restriction measures and vaccination.” It look very superficial conclusion. Let the readers know few important variables with their measure of strength that influences public trust.

6. PLOS authors have the option to publish the peer review history of their article (what does this mean?). If published, this will include your full peer review and any attached files.

**Do you want your identity to be public for this peer review?** For information about this choice, including consent withdrawal, please see our Privacy Policy.

Reviewer #1: No

Reviewer #2: No

---

## [Editor Report · Decision Letter 1]

19 Sep 2022

PGPH-D-22-00823R1

Influence of conspiracy theories and distrust of community health volunteers on adherence to COVID-19 guidelines and vaccine uptake in Kenya

Dear Dr. Ireri

Thank you for submitting your revised  manuscript to PLOS Global Public Health, however your response letter does not clearly state what changes you have made and how you have addressed the reviewers comments. Therefore, for us to consider your paper for publication, I invite you to submit a revised version of the manuscript with a response letter clearly outlining all changes and with point-by-point response to all the queries raised by the reviewers. 

Please submit your revised manuscript with response letter by 15 October 2022. If you will need more time than this to complete your revisions, please reply to this message or contact the journal office at globalpubhealth@plos.org. Please include the following items when submitting your revised manuscript:

We look forward to receiving your revised manuscript.

Kind regards,

Nusrat Homaira

Academic Editor
---

## [Decision Letter · Decision Letter 2]

8 Jan 2023

PGPH-D-22-00823R2

Influence of conspiracy theories and distrust of community health volunteers on adherence to COVID-19 guidelines and vaccine uptake in Kenya

Dear Edward Mugambi Ireri

Thank you for submitting your manuscript to PLOS Global Public Health. After careful consideration, we feel that it has merit but does not fully meet PLOS Global Public Health’s publication criteria as it currently stands. Therefore, we invite you to submit a revised version of the manuscript that addresses the points raised during the review process.

The reviewers saw the clear improvements between drafts, but there are still a number of substantive suggestions they have made. Please do present the raw data in addition to the adjusted odds as discussed in one review and further demographics as discussed by the other reviewer. I would strengthen the conclusion/recommendations as discussed and also tighten the background shared and organize the methods section as suggested. There are also a number of small word choices that should be adjusted for standardization, such as conspiracy instead of conspirator and "pay cut" as two words.

We look forward to receiving your revised manuscript.

Kind regards,

Megan Coffee, MD, PhD

Academic Editor

Journal Requirements:

Additional Editor Comments (if provided):

Reviewers' comments:

Reviewer's Responses to Questions

**Comments to the Author**

1. If the authors have adequately addressed your comments raised in a previous round of review and you feel that this manuscript is now acceptable for publication, you may indicate that here to bypass the “Comments to the Author” section, enter your conflict of interest statement in the “Confidential to Editor” section, and submit your "Accept" recommendation.

Reviewer #2: All comments have been addressed

Reviewer #3: (No Response)

2. Does this manuscript meet PLOS Global Public Health’s publication criteria? Is the manuscript technically sound, and do the data support the conclusions? The manuscript must describe methodologically and ethically rigorous research with conclusions that are appropriately drawn based on the data presented.

Reviewer #2: Yes

Reviewer #3: Yes

3. Has the statistical analysis been performed appropriately and rigorously?

Reviewer #2: N/A

Reviewer #3: No

4. Have the authors made all data underlying the findings in their manuscript fully available (please refer to the Data Availability Statement at the start of the manuscript PDF file)?

Reviewer #2: Yes

Reviewer #3: Yes

5. Is the manuscript presented in an intelligible fashion and written in standard English?

Reviewer #2: Yes

Reviewer #3: Yes

6. Review Comments to the Author

Reviewer #2: Much better draft now. A few comments for further improvemnet.

1) Avoid using p values when you present 95% CI. You can keep the p-value in the table but don't need to show this in abstract and any other text.

2) Raw data is still missing. You presented the Adjusted odds ratio which is fine but you also need to present the raw data (number and percentage in the table). For example in Table 2 .. "1) Trust government COVID-19 response" -- you presented few values [0.707, 1.071, 1.501] which need to clarify in table headings (is it odds ratio? If yes, please write in the table headings). The most important thing is that - you need to show how many people had trust on Government Covid-19 response and what percentage (n, and %). You can specify Big N for each variable separately or if same at the table legend/title. Then you can keep these odds ratio and p-value. People who don't have understanding on Odds ratio can still judge your findings based on the percentage. Please keep all these details in all four table (Table 1-4).

Reviewer #3: Abstract

The first two sentences in the abstract can be combined into one.

The rationale is weak and needs to be strengthened.

Who conducted the study should be mentioned explicitly.

How were the respondents selected?

The data analysis plan is missing in the abstract.

Background

The background needs to be shortened. The background section should discuss what knowledge the study will add to the existing literature.

Materials and methods

The ethical aspects of the study could be discussed at the end of the methods section.

The background of the data collectors and their training should be discussed explicitly.

The method section could be benefitted by following the chronological order

• Study site and population.

• Outline the setting in which the study was carried out.

• Mention the study participants.

• Design and sampling

• Describe the study design/approach

• Provide key operational definitions

• Outline sampling methods

• Data collection?

• Outline data collection approaches

• Outline any special laboratory materials, equipment, or reagents.

• Data analysis

• Which variables were included in the multiple logistic regression, and why.

Results:

Socio-demographics need more data. Age, sex, education, and years as volunteers could be discussed here.

“The inferential statistics were evaluated and interpreted using the confidence level at 95% as a baseline”-could go under the method section.

61.1% -could be spelled out.

“conspiration theories”-Did you mean conspiracy theories?

“Household head (adjOR= 0.409, 95% CI: 0.213-0.785; p = 0.007) and gender (adjOR= 2.780, 95% CI: 1.526-5.064; p = 0.001), were significantly associated with trust in the COVID-19 vaccination initiative”. The idea is not clear here. We need to discuss the category. For example, Male gender was significantly associated …………….

Discussion:

The authors need to use more references to support the statements discussed.

“A good example was the allegation that the Nyanza region immigration officials had issued 25 foreigners from India to work at Kibos sugar and allied industries in Siaya county. Five of them tested positive for the delta variant of COVID-19” A reference is needed.

Page 18-foot noted could be deleted and used as a reference.

younger CHV—Age is needed here

The authors need to discuss the limitations of the study and focus on the impact of these limitations have on the conclusions.

Recommendations need to be strengthened. The authors could discuss the next steps that are practical and applicable to the context. What should specific research questions next be pursued?

7. PLOS authors have the option to publish the peer review history of their article (what does this mean?). If published, this will include your full peer review and any attached files.

**Do you want your identity to be public for this peer review?** For information about this choice, including consent withdrawal, please see our Privacy Policy.

Reviewer #2: No

Reviewer #3: **Yes: **Md Saiful Islam

---

## [Decision Letter · Decision Letter 3]

27 Feb 2023

Influence of conspiracy theories and distrust of community health volunteers on adherence to COVID-19 guidelines and vaccine uptake in Kenya

PGPH-D-22-00823R3

Dear Edward Mugambi Ireri

We are pleased to inform you that your manuscript 'Influence of conspiracy theories and distrust of community health volunteers on adherence to COVID-19 guidelines and vaccine uptake in Kenya' has been provisionally accepted for publication in PLOS Global Public Health.

Best regards,

Megan Coffee, MD, PhD

Academic Editor

Reviewer Comments (if any, and for reference):

Reviewer's Responses to Questions

**Comments to the Author**

1. If the authors have adequately addressed your comments raised in a previous round of review and you feel that this manuscript is now acceptable for publication, you may indicate that here to bypass the “Comments to the Author” section, enter your conflict of interest statement in the “Confidential to Editor” section, and submit your "Accept" recommendation.

Reviewer #2: All comments have been addressed

Reviewer #3: All comments have been addressed

2. Does this manuscript meet PLOS Global Public Health’s publication criteria? Is the manuscript technically sound, and do the data support the conclusions? The manuscript must describe methodologically and ethically rigorous research with conclusions that are appropriately drawn based on the data presented.

Reviewer #2: Yes

Reviewer #3: Yes

3. Has the statistical analysis been performed appropriately and rigorously?

Reviewer #2: Yes

Reviewer #3: Yes

4. Have the authors made all data underlying the findings in their manuscript fully available (please refer to the Data Availability Statement at the start of the manuscript PDF file)?

Reviewer #2: Yes

Reviewer #3: Yes

5. Is the manuscript presented in an intelligible fashion and written in standard English?

Reviewer #2: Yes

Reviewer #3: Yes

6. Review Comments to the Author

Reviewer #2: The authors addressed most of the comments ! Very well done!

Reviewer #3: Accept

7. PLOS authors have the option to publish the peer review history of their article (what does this mean?). If published, this will include your full peer review and any attached files.

**Do you want your identity to be public for this peer review?** For information about this choice, including consent withdrawal, please see our Privacy Policy.

Reviewer #2: **Yes: **Najmul Haider

Reviewer #3: **Yes: **Md Saiful Islam
